# Auranofin Has Advantages over First-Line Drugs in the Treatment of Severe *Streptococcus suis* Infections

**DOI:** 10.3390/antibiotics10010026

**Published:** 2020-12-30

**Authors:** Hao Lu, Wenjia Lu, Yongwei Zhu, Chenchen Wang, Liming Shi, Xiaodan Li, Zhaoyuan Wu, Gaoyan Wang, Wenqi Dong, Chen Tan, Manli Liu

**Affiliations:** 1State Key Laboratory of Agricultural Microbiology, College of Veterinary Medicine, Huazhong Agricultural University, Wuhan 430070, China; 88251420@webmail.hzau.edu.cn (H.L.); 2017302110131@webmail.hzau.edu.cn (W.L.); zhuyongwei@webmail.hzau.edu.cn (Y.Z.); 2018302110164@webmail.hzau.edu.cn (C.W.); xiaodanLi@webmail.hzau.edu.cn (X.L.); 97wgy@webmail.hzau.edu.cn (G.W.); dongwq@webmail.hzau.edu.cn (W.D.); 2Heze Municipal Hospital, Heze 274000, China; hzslyykjk@163.com; 3Hubei Biopesticide Engineering Research Centre, Wuhan 430070, China; zhaoyuan.wu@nberc.com; 4Key Laboratory of Preventive Veterinary Medicine in Hubei Province, The Cooperative Innovation Center for Sustainable Pig Production, Wuhan 430070, China; 5Key Laboratory of Development of Veterinary Diagnostic Products, Ministry of Agriculture of the People’s Republic of China, Wuhan 430070, China; 6International Research Center for Animal Disease, Ministry of Science and Technology of the People’s Republic of China, Wuhan 430070, China

**Keywords:** STSLS, *Streptococcus suis*, auranofin, molecular docking

## Abstract

Streptococcal toxic shock-like syndrome (STSLS) likely occurs when an individual is infected with the *Streptococcus suis* (*S. suis*) epidemic strain and is characterized by a cytokine storm, multiple organ dysfunction syndrome (MODS) and a high incidence of mortality despite adequate treatment. A number of antibiotics exhibit excellent bactericidal effects in vivo, such as fluoroquinolones, aminoglycosides (gentamicin) and β-lactams (penicillin G, ceftiofur, or amoxicillin), but are less effective for treating STSLS. Therefore, there is an urgent need to identify new compounds that can reduce the damage caused by STSLS. In the present study, we identified auranofin, an orally bioavailable FDA-approved anti-rheumatic drug as a candidate repurposed drug to treat severe *S. suis* infections. Our results showed that auranofin can bind to the functional domain of bacterial thioredoxin reductase, decreasing the reducing redox-responsive capacity of target bacteria and allowing for the killing of *S. suis* cells. We also observed that auranofin has antibacterial activity against other gram-positive bacteria, such as multidrug resistant Streptococcus pneumoniae (MDRSP), Streptococcus agalactiae, and vancomycin-resistant strains of Staphylococcus aureus. Additionally, auranofin is capable of eradicating intracellular *S.suis* present inside infected macrophage cells. Mouse model experimental results showed that auranofin could effectively reduce the mortality of mice infected with *S. suis*. Compared to the ampicillin treatment group, the survival rate of mice in the auranofin treatment group in severely infected model mice was significantly improved. These results suggest that auranofin has the potential for use as an effective antibiotic against *S. suis.*

## 1. Introduction

*Streptococcus suis* (*S. suis*) can cause many symptoms, such as septicemia, meningitis, arthritis, and endocarditis in both humans and pigs and is associated with high mortality [1,2,3]. Based on incomplete data, since the first case was described in Denmark in 1968, over 1600 human cases of *S. suis* infection have been reported [4,5]. In Asia, this pathogen affects the general population and is of significant public health concern [6]. Among all the known *S. suis* serotypes, *S. suis* serotype 2 (SS2) is the most prevalent in both pigs and humans and is often reported worldwide [7]. *S. suis* is also an important pathogen that causes meningitis in Vietnam, Thailand, and Hong Kong [8]. During July and August of 2005, a sudden outbreak of 215 human cases *S. suis* infection was described in Sichuan province, where Streptococcal toxic shock-like syndrome (STSLS) caused by this bacterium was reported to cause severe symptoms, such as acute high fever, hypotension, shock, blood spots, and dysfunction of multiple organs. This outbreak was also associated with acute death and a mortality rate higher than 80%, even after adequate treatment [9].

Inflammatory cytokines storms are one of the leading causes of death from many diseases, such as the SARS-CoV1 epidemic in 2003 and the COVID-19 pandemic this year, which can rapidly progress to cytokine storm syndrome, organ dysfunction, and death [10]. The characteristics of STSLS include a high bacterial burden, an inflammatory cytokine storm, multiple systemic organ failure, and eventually, acute death of the host [10]. A clinical retrospective study showed that in the blood of patients with STSLS, the levels of interleukin (IL)-1β, IL-6, IL-8, IL-12, tumor necrosis factor-alpha (TNF-α), and interferon-γ(IFN-γ) are much higher than those of normal individuals [10]. For STSLS, the results of further studies showed that the induction of an inflammatory cytokine storm is essential [11,12]. It is worth noting that the level of the inflammatory response and organ damage caused by *S. suis* 2 strain SC19 is much higher than the classical virulent P1/7 *S. suis* strain, which could also cause high mortality [10]. Therefore, inhibiting the inflammatory cytokine storm caused by *Streptococcus suis* is key to curing streptococcicosis.

Some antibacterial drugs function by adversely affecting metabolic and homeostatic networks [13]. One basic biological pathway is thiol-based redox metabolism, which is essential for many cellular processes [14]. The *S. suis*’s Trx reductase (TrxB) are part of the the thioredoxin system and appear to be essential for *S.suis* [15,16]. Auranofin is an oral gold-containing drug initially approved by the U.S. Food and Drug Administration (FDA) for treatment of rheumatoid arthritis. More recent studies have discovered auranofin exerts antibacterial activity against a variety of pathogens [17,18,19,20,21]. At present, whether the thioredoxin reductase is the sole target of auranofin in bacteria remains controversial [19,22]. In the present study, we demonstrated that auranofin can bind to TrxB protein active centers to inhibit their function, with the resulting increase in reactive oxygen species (ROS) levels being the primary cause of *S. suis* cell death. Even more remarkable is that we showed that auranofin significantly improved the survival rate of severely infected model mice compared to mice treated with the clinically common drug ampicillin. These findings suggest that auranofin may be a promising therapeutic candidate for the treatment of *S. suis* infection.

## 2. Results

### 2.1. Auranofin Shows Excellent Antibacterial Activity

To assess the antimicrobial activity of auranofin, we evaluated the effect on auranofin against 18 multidrug-resistant Staphylococcal and Streptococcal strains (Table 1). Auranofin showed significant growth inhibition activity against all tested strains at a low working concentration (0.0625–0.25 µg/mL), even those pathogens that were resistant to commonly used antimicrobials, such as ampicillin, tetracycline, levofloxacin, cefotaxime, and imipenem. In addition to possessing anti-*S. suis* activity, auranofin also showed strong antibacterial activity against multidrug resistant *S. pneumoniae* (MDRSP), *S. agalactiae* and vancomycin-resistant *S. aureus* (VRSA), with minimum inhibitory concentration (MIC) values ranging from 0.0625–0.125 μg/mL, suggesting that the development of cross-resistance between auranofin and these antibiotics is unlikely to occur. Antimicrobial compounds are typically defined as bactericidal if the minimum bactericidal concentration (MBC)does not exceed 4× the MIC [23]. To further to assess the effect of auranofin against SC19, a time-kill assay was performed using auranofin at a concentration of 4× MIC. The results showed that the CFU decreased by at least 3 logs after a 12 h incubation (Figure 1).

### 2.2. Safety Evaluation of Auranofin

Although auranofin is an FDA-approved drug that has been in clinical use for several decades, we conducted a number of assays to evaluate the toxicity of auranofin. To this end, Vero or RAW264.7 cells were treated with different concentrations of auranofin (0.25–32 µg/mL) and then assessed for viability. At an auranofin concentration of 32 µg/mL, the viability rate of Vero cell was 50% and rate of RAW264.7 cell was 60% revealing that the toxicity of auranofin toward cells was much higher than the MIC against *S. suis* (Figure 2A). In addition, in vivo toxicity test at 20 times therapeutic dose, no differences in the liver and kidney blood indices of urea nitrogen, creatinine, ALT, and AST were observed between the medicine group (auranofin at the dose of 2.4 mg/kg/day) and the control group (Table 2), indicating that severe damage was not induced by auranofin treatment. Furthermore, no weight loss or other obvious comorbidities were observed in the treated mice (Figure 2B).

### 2.3. Mechanism of Auranofin Antibacterial Activity

To identify the binding site of the auranofin in TxrB, we used the molecular docking method, and the estimated binding energy was determined as −5.5 kcal/mol. Then, the hypothetical binding mode of auranofin in the binding site of TxrB was determined. In this model, we observed that auranofin adopted a compact conformation to bind the binding site of TxrB (Figure 3A). In addition, auranofin stretched into the hydrophobic pocket of the TxrB binding site that consisted of Cys-130, Ala-131, Val-132, Phe-158 and Leu-279, forming a strong hydrophobic structure. Importantly, one key hydrogen bond interaction was observed with a bond length of 3.3 Å was observed between auranofin and the Glu-155 residue of TxrB that was the primary interaction between these molecules (Figure 3B). All of these interactions promoted the anchoring of auranofin in the TxrB binding site. The interaction between auranofin and TrxB was also assessed based on isothermal titration calorimetry (ITC) tests. We observed an equilibrium dissociation constant (KD) of 2.136 × 10^−6^ mol/L (Figure 3C) between auranofin and TrxB, suggesting that auranofin has a high affinity for TrxB.

### 2.4. Auranofin Alleviates the SC19-Induced Injury of RAW264.7 Cells

Although *S. suis* is an atypical intracellular bacterium, the SS2 virulence factor SLY has been reported to promote host cell perforation [24]. *S. suis* is capable of entering different types of cells, including macrophages, in mammalian tissues, allowing it to evade host defenses and for infections to last an extended period of time. Such infections are particularly challenging, because many antibiotics are unable to penetrate the cell membrane and enter the intracellular niche to kill the bacteria [25,26]. In the present study, uninfected RAW264.7 cells displayed green fluorescence when stained with a LIVE/DEAD (green/red) staining reagent (Figure 4A). In contrast, after infection with SC19 for 1 h, an increased number of RAW264.7 cells with red fluorescence was observed (Figure 4B), indicating that SC19 caused macrophage injury and death. Interestingly, the addition of auranofin (0.25 mg/L) protected RAW264.7 cells from SC19-mediated cell injury, as demonstrated by the significant reduction in red fluorescence (Figure 4C). To quantify the protective effect of auranofin in clearing intracellular SC19, its activity was tested against RAW264.7 macrophages infected with SC19. Auranofin effectively cleared more than 82% of intracellular *S. suis* cells (Figure 4D) at a nontoxic concentration of 0.25 μg/mL. These results suggest that auranofin can kill *S. suis* cells harbored by macrophages. These findings suggest that auranofin is a potential valuable treatment option for challenging infections/diseases (such as pneumonia) in which *S. suis* resides within host cells.

### 2.5. Auranofin Treatment of SC19 Cells Results in Thiol Depletion and Compromises Their Defense against Oxidative Stress

To test whether TrxB levels are impacted antibacterial treatment, we assayed for the thiol content of both SC19 and pSET2-TrxB/SC19 cells treated with auranofin or ampicillin. Indeed, *S. suis* treated with auranofin at 0.5× or 1× MIC showed 22 and 35% decreases in the amount of cellular free thiols, respectively (Figure 5A). Only a slight decrease in free thiols was observed in pSET2-TrxB/SC19 cells (Figure 5B), and no change was observed when these cells were treated with ampicillin (Figure 5A,B). These results suggest that bacterial thiol-redox homeostasis would be disrupted by auranofin-mediated inhibition of TrxB. The loss of cell reduction also inhibits bacterial defenses against reactive oxygen species, including the activities of a variety of thiol-dependent enzymes [27]. As previously reported, ROS is crucial for bactericidal agents [28], and in the present study, the decrease in free thiols stimulated the accumulation of ROS (Figure 5C), further disrupting bacterial homeostasis. Subsequently, we evaluated the growth performance of the TrxB overexpression strain pSET2-TrxB/SC19 in the presence of 2 mg/L of auranofin. The pSET2-TrxB/SC19 strain showed notably increased resistance to auranofin compared to SC19 cells (Figure 5D). These results suggest that auranofin has antibacterial effects by severely damaging the defense of bacteria against oxidative stress. Furthermore, we used paraquat, which generates intracellular ROS, as a positive control to assess the antimicrobial activity of auranofin. Paraquat alone showed little antimicrobial activity against *S. suis* (0.5-log decrease in CFU at 5 mM), but when combined with 0.25 mg/L of auranofin, the two compounds exhibited notable synergism (a ~4.2-log decrease in CFU) (Figure 5E).

### 2.6. Protective Rates of Auranofin and Ampicillin in Infected Mouse Models at Different Times

The efficacy of auranofin was evaluated in severely infected mouse models at different infection times. At 1 or 6 h post infection (hpi) with SC19, mice (10 mice per group) were orally administered auranofin or ampicillin, while mice in the untreated group received an equal volume of PBS. Subsequently, the mortality rates of the mice in each group were observed for 7 days. The groups treated with auranofin or ampicillin exhibited a survival rate of 90% at 1 hpi, indicating that auranofin and ampicillin had a good treatment effect. However, the mice in the group treated with auranofin displayed the survival rate of 70% at 6 hpi, while that observed in the group treated with ampicillin was only 20% (Figure 6A). These results showed that auranofin exhibited a much higher protection rate against severe infection in mice than ampicillin, a first line drug.

### 2.7. Anti-Inflammatory Effects of Auranofin Are Crucial for Improving the Rate of Protection

We further assessed the mechanism associated with the difference in protection rates observed between the two drugs. The CFU determined for the groups treated with the two drugs at different time points were significantly lower than that of the untreated group (Figure 6B–E), suggesting that the ability of auranofin to kill bacteria in vivo was similar to that of ampicillin therapies. The untreated group (1 h post infection model) showed significant inflammatory responses, whereas neither ampicillin nor auranofin showed a significant increase in inflammatory cytokines (Figure 7A). These results suggest that STSLS can be prevented by prompt drug therapy during early infection. However, the group treated with ampicillin at 6 hpi showed a significant inflammatory response that was similar to that observed in the untreated group, whereas the level of inflammation in the auranofin treatment group was approximately 70% lower than that observed in the ampicillin treatment group (Figure 7A). Thus, auranofin exhibited an advantage over ampicillin in reducing the inflammatory response, since the latter tended to only have a bactericidal effect in severely infected mouse models. Subsequent blood biochemistry results further confirmed this observation. The group (6 h post infection model) treated with ampicillin exhibited SC19-induced inflammatory storms that caused multiple organ damage, which was similar to that observed in the untreated group. In these two groups, the levels of ALT, AST, and CK were significantly increased. However, in the auranofin treatment group (6 h post infection model), AST and CK levels were increased slightly (Figure 7B). Furthermore, H&E staining results indicated that auranofin treatment alleviated inflammatory manifestations, including inflammatory cell infiltration, pulmonary vessel dilatation, alveolar interstitial congestion, and edema in severe infection mouse models (Figure 7C). These results may explain why auranofin and ampicillin exhibited similar effects in the early stages of infection, as both drugs exhibited good in vivo antibacterial ability and at the same early stage of *S. suis* infection, when a strong immune response is not triggered. However, in the case of severe infection, antibacterial treatment with ampicillin may not reduce the inflammatory damage to the body. In contrast, auranofin had a better effect than the first-line drug ampicillin due to its dual anti-inflammatory and bactericidal effects, providing a higher protection rate.

## 3. Discussion

Auranofin has been used for medicinal purposes for centuries. First approved as an oral gold therapy in 1985, auranofin is one of only three gold complexes currently approved for clinical use. The results of previous studies have confirmed the safety of auranofin and revealed its pharmacokinetic profile in humans, laying a good foundation for further investigation of auranofin in clinical application.

Auranofin exhibits excellent antibacterial activity against multidrug-resistant *Streptococcus suis*, Streptococcus pneumonia (MDRSP), Streptococcus agalactiae, and vancomycin-resistant Staphylococcus aureus (VRSA) at low inhibitory concentrations (0.0625–0.25 mg/L), much lower than the achievable drug concentration in human plasma (2.37 mg/L) and indicating that auranofin is a promising drug [18]. At present, whether TrxB is the primary target of auranofin in gram-positive bacteria is controversial [22,29]. In the present study, we performed a standard molecular simulation for the TrxB-auranofin complex. Based on MD simulation and binding free energy calculations, we observed that Morin can bind the 2 domain of SLY by forming strong contact with residues of Thr49, Tyr54, Gln107, Asn50, and Asp111. These results were confirmed by ligand-residue interaction decomposition using the MM-PBSA method, residue point mutations, and a fluorescence-quenching assay. Further ITC results demonstrated strong binding between TrxB and auranofin. Subsequently, we constructed the TrxB overexpression strain pSET2-TrxB/SC19 and observed that the MIC of pSET2-TrxB/SC19 increased 4-fold compared to the wild-type strain. These results demonstrated that TrxB is the primary target of auranofin in *S. suis*.

Auranofin is approved for a long-term daily dosage of 6 mg/day and rarely causes severe side effects, with the most common symptom being gastrointestinal distress, which is easily cured [30]. Patients treated with auranofin have been monitored in clinical trials for longer than 5 years, a larger interval than expected for normal antibiotic treatment and have shown no cumulative toxicity. In our present study, we tested mice at a 20-fold therapeutic dose and observed no significant changes in biochemical parameters or body weight. Auranofin showed favorable in vivo activity against multidrug-resistant clinical isolates of *S. suis*. Specifically, the severe infection model showed a higher protection rate than commonly used drugs. Previous studies have shown that the activation of NLRP3 is the main cause of STSLS [29]. It has been proved that auranofin can play an anti-inflammatory role by inhibiting NLRP3 [31,32]. Our results show that for some severely infected patients, auranofin alone or in combination with first-line drugs may be a better choice. Furthermore, auranofin showed the ability to clear intracellular *Streptococcus suis* from infected macrophages. These findings suggest that auranofin is a potentially valuable therapeutic option for intracellular infections/diseases. What’s more, as auranofin is an approved and off-patent drug, it could be quickly and economically tested in clinical trials and, if successful, in patients. Thus, auranofin is a drug that deserves further study.

## 4. Materials and Methods

### 4.1. Bacterial Strains, Growth Conditions, Auranofin Preparation

*Streptococcus suis* serotype 2 (SS2) strain SC19 used in the present study is a virulent strain isolated from the brain of a dead pig in Si Chuan province of China in 2005 [33]. The TrxB overexpression vector pSET2-TrxB/SC19 was constructed using the Escherichia coli-*Streptococcus suis* shuttle vector pSET2 [34]. Relevant information for the bacterial strains and used in the present study is listed in Table 1. Trypticase soy broth (TSB), trypticase soy agar (TSA), were purchased from Difco Laboratories, Detroit, MI, USA. Auranofin was purchased from Topscience, and a 0.22-µm syringe filter (Millipore, Burlington, MA, USA) was used to filter auranofin dissolved in dimethyl sulfoxide (Sigma-Aldrich, Burlington, MA, USA) to make stock solutions of various concentrations.

### 4.2. Assessment of the Anti-S. suis Activity of Auranofin

The minimum inhibitory concentration of auranofin against strain was determined following the Clinical And Laboratory Standards Institute (CLSI) guidelines [35]. The microdilution broth method was performed in 96-well plates (Corning Costar^®^ 3599 Corning, Corning, NY, USA) using MHB (Hopebio, Qingdao, China). The final concentration of the culture was 5 × 10^5^ colony-forming units (CFU)/mL, and measurements were repeated at least in triplicate. The measurements were repeated in triplicate.

### 4.3. Time-Kill Curve

To further evaluate the antibacterial activity of auranofin, a time-kill curve was generated. Overnight cultures of bacteria were subcultured in TSB and then diluted to 106 cells/mL in MHB supplemented with 5% fetal bovine serum. Then, auranofin was added at a concentration of 4× MIC, and the mixture was incubated at 37 °C. After different time intervals, bacterial suspensions were plated onto TSA, and viable bacterial cells were counted. The measurements were repeated in triplicate.

### 4.4. Cell Toxicity Test

A WST-8(US Everbright^®^ Inc., Suzhou, China) assay was performed after Vero or RAW264.7 cells were seeded at a density of approximately 40,000 cells per well in a 96-well tissue culture plate. Then, auranofin was added to the appropriate wells at final concentrations of 0.25–32 μg/mL, and the cells were incubated for 24 h. Subsequently, the cells were washed with PBS three times, and the WST-8 assay reagent was added. After a 4 h incubation at 37 °C under an atmosphere with 5% CO_2_, the absorbance of each well was measured at 490 nm using a FLUOstar Omega instrument. The measurements were repeated in triplicate.

### 4.5. In Vivo Toxicity Experiment

To evaluate the toxicity of auranofin toward major organs, such as the liver and kidney, mice were randomly divided into the PBS control and treatment groups (5 mice in each group). The mice in the treatment group were orally administered 2.4 mg/kg/d of auranofin (20 times the therapeutic dose) for three days. Then, 12 h after the last administration, blood samples were collected from the anesthetized animals. Biochemical analysis was performed using an automatic analyzer (Chemray 800, Shenzhen, China), with alanine aminotransferase (ALT), aspartate aminotransferase (AST), creatinine, and urea nitrogen levels analyzed. The weights of the mice were assessed over seven days [36].

### 4.6. Production of Recombinant TrxB Protein

The TrxB gene was PCR amplified from SC19 genomic DNA with the primers P1 and P2 (listed in Table 3). The gene TrxB was subcloned into the HindIII and XhoI restriction enzyme sites of the vector pET-28a (+) (Novagen, Madison, WI, USA) with to construct the prokaryotic expression plasmid pET-28a (+)-TrxB. Then, the recombinant plasmid encoding the histidine-tagged TrxB protein (rTrxB) was transformed into *E. coli* BL21 (DE3). Recombinant protein expression was induced by the addition of 0.5 mM isopropyl-β-d-thiogalactopyranoside (IPTG) at 16 °C for 20 h. Subsequently, the supernatant of bacterial cell lysates was loaded onto a Ni-NTA column to purify the rTrxB protein.

### 4.7. Homology Modeling and Molecular Docking

The amino acid sequence of the *S. suis* TxrB protein was obtained using the NCBI protein database (http://www.ncbi.nlm.nih.gov/protein/) and had the accession number NZ_LS483418.1. The BLAST server (http://blast.ncbi.nlm.nih.gov) was used for searches using the protein template. We used Streptococcus pyogenes thioredoxin reductase (PDB ID: 5 VEU) as TxrB template to compare the homology of its amino acid sequence. TxrB homology modeling was performed using SWISS-MODEL (https://swissmodel.expasy.org/), a fully automated protein structure homology modeling server. Autodock Vina 1.1.2 [37] was used to perform molecular docking of TxrB and auranofin, which improved the speed and accuracy of the docking through a novel scoring function. ChemBioDraw Ultra 14.0 and ChemBio3D Ultra 14.0 (Cambridge Inc., Bedford, OH, USA) were used to draw the 2D and 3D structures of auranofin, and the AutoDockTools 1.5.6 package (The Scripps Research Institute, La Jolla, CA, USA) was used to generate the docking input files [38,39].

### 4.8. Auranofin and TrxB Binding Assays

Auranofin binding was assessed using isothermal titration calorimetry at 25 °C with a NANO-ITC (TA Instruments, New Castle, TA, USA). Solutions of the purified TrxB protein and auranofin were prepared at a concentration of 0.02 0.2 mmol/L in PBS (pH 7.4). The auranofin was added 20 times (every 2.5 μL) into the protein solution (volume = 300 µL) with equilibration intervals of 200 s. The obtained data were processed using the software with the instrument to calculate the equilibrium dissociation constant (KD).

### 4.9. Construction of TrxB-Overexpressing Strains

The TrxB gene was amplified from SC19 genomic DNA using the primers listed in Table 2 and then ligated into the plasmid pSET2 to construct the overexpression plasmid pSET2-TrxB. Subsequently, the plasmid was transferred into SC19 to generate the overexpression strain.

### 4.10. Thiol Depletion Assay

*S. suis* cells were treated with the appropriate concentrations of auranofin or ampicillin for 15 min. Subsequently, the cells were washed three times with PBS and then resuspended in 100 mM potassium phosphate (monobasic) containing 1 mM ethylenediaminetetraacetic acid (pH 7.4) and lysed using a Precellys 24 homogenizer (Bertin Corp, 2096 Gaither Rd, Rockville, MD, USA). Thiol depletion was determined using a Thiol Detection Assay kit (Cayman Chemical, Ann Arbor, MI, USA).

### 4.11. ROS Measurement

The ROS content in *S. suis* cells was measured with 10 µmol/L of DCFH-DA, following the manufacturer’s instructions, with some modifications (Beyotime, Shanghai, China) [40]. Briefly, *S. suis* cells were grown overnight at 37 °C, washed and then adjusted to a density at OD600 of 0.5 in PBS. Then, the cells were incubated in DCFH-DA at 37 °C for 30 min. Subsequently, after washing the cells three times, 190 µL of the bacterial cell suspension was added to a black 96-well plate and mixed with 10 µL of auranofin (10× MIC). After another 30 min incubation, the fluorescence intensity of the sample was immediately measured with excitation and emission wavelengths of 488 and 525 nm using a fluorescence microplate reader (SPARK 10M, TECAN, Männedorf, Switzerland).

### 4.12. Cell Culture and Infection

RAW264.7 mouse macrophage-like cells were cultured at 37 °C under an atmosphere with 5% CO_2_ in DMEM (Invitrogen) supplemented with 10% fetal bovine serum (FBS). The cells were incubated overnight in 96-well plates at a density of 2 × 10^4^ cells per well, after which they were infected with SC19 at OD600 = 0.8 and resuspended in FBS-free DMEM medium (MOI = 10) for 1 h. Subsequently, the cells were washed with PBS three times and then resuspended in fresh DMEM containing auranofin (0.25 mg/L) or ampicillin (0.5 mg/L) for 24 h. Finally, the cells were treated with 0.1% Triton X-100, diluted with PBS at a 1:10 ratio, and ten aliquots were spread onto TSA plates for colony enumeration. Microscopic images of stained cells were obtained using live/dead (green/red) reagents (Invitrogen) under a confocal laser scanning microscope (Nikon, Tokyo, Japan).

### 4.13. Animal Experiments

Seven-week-old female BALB/c mice were purchased from China Three Gorges University to establish mouse models of *S. suis* Sc19 infection. Animal experiments conformed to animal ethical guidelines, and all experiments were conducted under the guidance of the Protection, Supervision, and Control Committee of Animal Experiments of Huazhong Agricultural University (HZAUMO-2019-036). The animal experiments were performed as previously described with some modifications [22,41]. Five groups of 10 mice were intraperitoneally injected with 200 µL of SC19 cells (2.5 × 10^9^ cells/mL). At 1 or 6 h pi, the mice were orally administered auranofin (0.12 mg/kg/day) or ampicillin (20 mg/kg/day) for 3 days, which is equivalent to commonly used dose in human clinical practice. Mice in the untreated group (10 mice in each group) administered the same amount of PBS. The mortality rate of mice was observed for 7 days after treatment.

Another 6 groups of mice (5 in each group) were intraperitoneally injected with the same amount of bacterial suspension following the same process described above. The mice in the control group were only injected with PBS. Blood samples were collected from the anesthetized animals eight hours after the drug treatment and analyzed for the levels of alanine aminotransferase (ALT), aspartate aminotransferase (AST) and creatine kinase (CK). The infected tissues were ground, diluted, and plated onto TSA plates containing 10% bovine serum, and the CFU in the lungs, spleen, brain, and liver were enumerated. Cytokines were quantified via flow cytometry bead arrays (BD Biosciences, New York, NY, USA). The left lung of each mouse was fixed with 4% paraformaldehyde to observe the pathological changes caused by bacteria. At the end of the experiment, all the mice were euthanized by CO_2_ inhalation. Cytokines were quantified using a sensitive platform based on electrochemical luminescence (Quickplex, Meso-Scale Discovery^®^, Kenilworth, MD, USA) [42]. To assess the pathological changes caused by bacteria, the left lobe of each mouse lung was fixed in 4% paraformaldehyde for pathological examination. All mice were euthanized by CO_2_ inhalation. The cytokine assay was repeated 3 times.

### 4.14. Statistical Analysis

All experimental data (*n* ≥ 3) are expressed as the means ± SD. GraphPad Prism 8.0.2 (GraphPad Software, San Diego, CA, USA) was used for statistical analysis using two-tailed unpaired *t*-test.

### 4.15. Ethical Approval

All animal experimental schemes and operating techniques have been approved by the Animal Experiment Protection, Supervision and Control Committee of Huazhong Agricultural University (HZAUMO-2019-036) with strict reference to the Regulations of Hubei Province on the Administration of Experimental Animal Affairs and the Regulations of China on the Administration of Experimental Animal Affairs.

## 5. Conclusions

In the present study, we identified auranofin as a candidate repurposed drug to treat severe *S. suis* infections. Our results showed that auranofin exhibited a much higher protection rate against severe infection in mice than ampicillin, a first line drug, and reduced the STSLS in severe *S. suis* infections. More importantly, auranofin Inhibited the level of inflammatory cytokine in *S. suis* infection mouse, which was the key to reduce the mortality rate of *S. suis*. In conclusion, our findings suggested that auranofin could be a potential compound for treating severe *S. suis* infection.

## Figures and Tables

**Figure 1 antibiotics-10-00026-f001:**
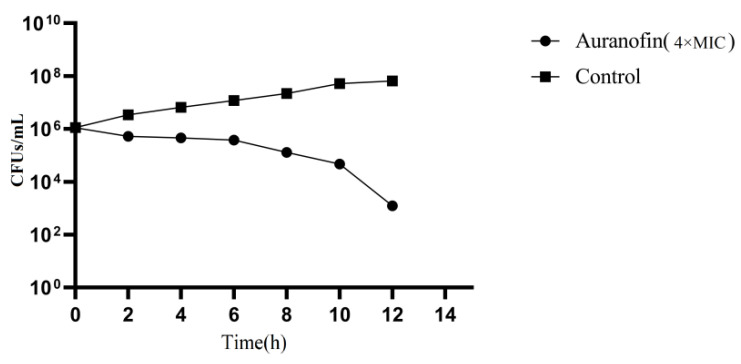
CFU of SC19 treated with DMSO or auranofin at a concentration of 4× MIC (MIC value: auranofin 0.125 mg/L).

**Figure 2 antibiotics-10-00026-f002:**
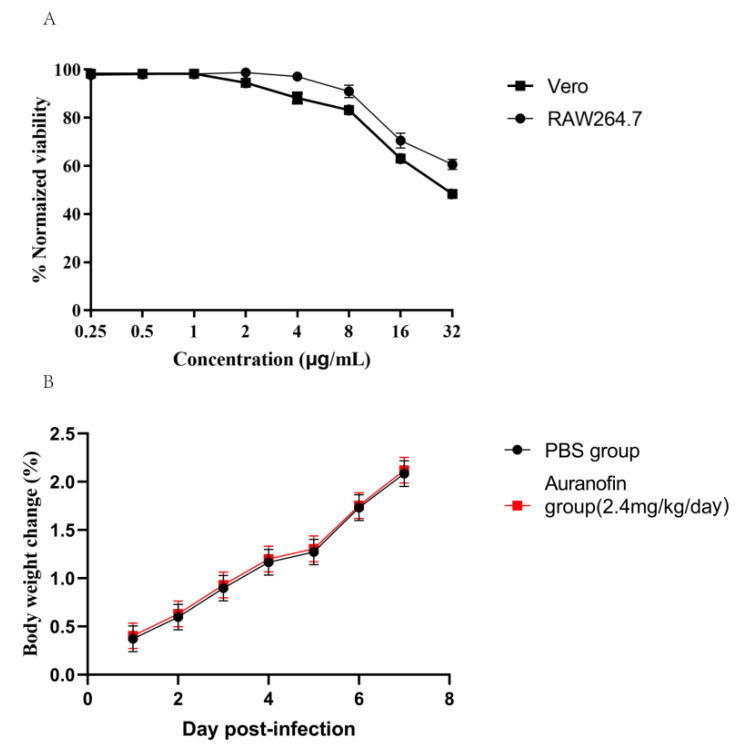
(**A**) Vero or RAW264.7 cell viability was measured after treatment with serially dilutions of auranofin (0.25–32 mg/L). (**B**) Body weight changes of BALB/c mice treated with auranofin or PBS. The average weight change of mice was calculated daily. The error bars represent the standard deviations of at least three independent experiments.

**Figure 3 antibiotics-10-00026-f003:**
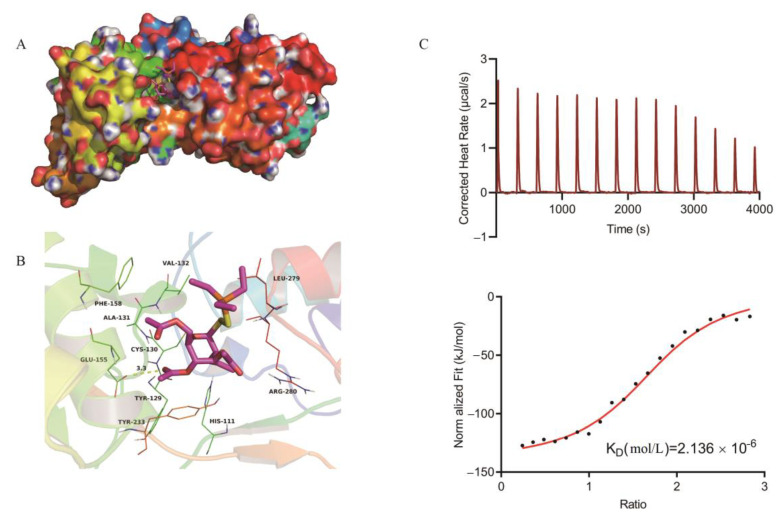
Auranofin can disrupt thiol−redox homeostasis by directly binding to TrxB. (**A**) Auranofin was docked into the binding site of TrxB (overall view). (**B**) The auranofin and TxrB binding site (detailed view). The representative binding residues within 4.0 Å of this substrate are shown in lines; auranofin is represented with red sticks; the hydrogen bond is shown as a yellow dotted line. (**C**) ITC analysis of the interaction between auranofin and TxrB, where 0.2 mmol/L of auranofin (50 μL) was added to 0.02 mmol/L of TrxB (300 μL) at 25 °C. Thermodynamic parameters were calculated the equilibrium dissociation constant (KD = 2.136 × 10^−6^ mol/L).

**Figure 4 antibiotics-10-00026-f004:**
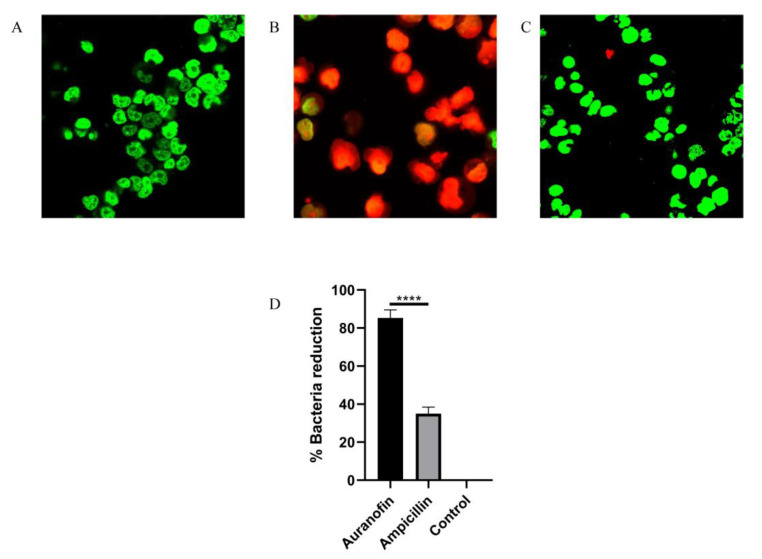
Auranofin protects RAW264.7 cells from SC19-mediated cell damage. (**A**) SC19-infected cells, Scale bar, 10 µm. (**B**) untreated cells + auranofin, Scale bar, 10 µm. (**C**) SC19-infected cells treated with 0.25 mg/L of auranofin. Scale bar, 10 µm (**D**) SC19-infected RAW264.7A.1 cells were treated with auranofin or ampicillin for 24 h, and the percent bacterial reduction was calculated compared to the uninfected control groups. The results are presented as the means ± SD (*n* = 3), **** *p* < 0.0001.

**Figure 5 antibiotics-10-00026-f005:**
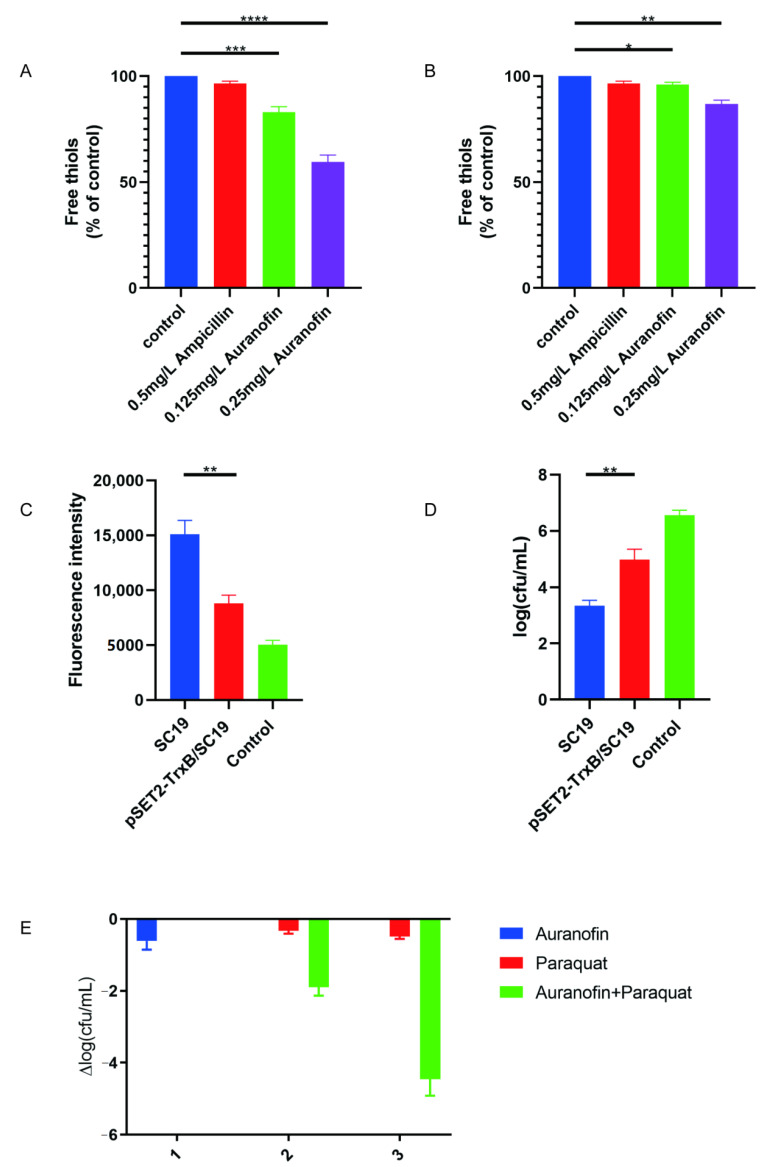
Auranofin depletes intracellular thiols and sensitizes *S. suis* to oxidizing agents and oxidative stress. (**A**) The free thiol concentration in *Streptococcus suis* cells was significantly decreased after auranofin treatment compared to that observed in the control group. (**B**) PSET2−TrxB/SC19 cultures treated with auranofin only show a slight dose−dependent depletion of thiols. (**C**) The ROS content in SC19 or pSET2−TrxB/SC19 cells treated with 2.5 mg/L auranofin. (**D**) PSET2−TrxB/SC19 showed notably increased resistance to auranofin compared to the SC19 strain. (**E**) The combination treatment of *S. suis* with auranofin and paraquat exhibited synergistic antimicrobial activity. *S. suis* cultures were treated for 4 h with the indicated concentrations of paraquat and 0.25 mg/L auranofin, separately or combination. * *p* < 0.05; ** *p* < 0.01; *** *p* < 0.001; **** *p* < 0.0001.

**Figure 6 antibiotics-10-00026-f006:**
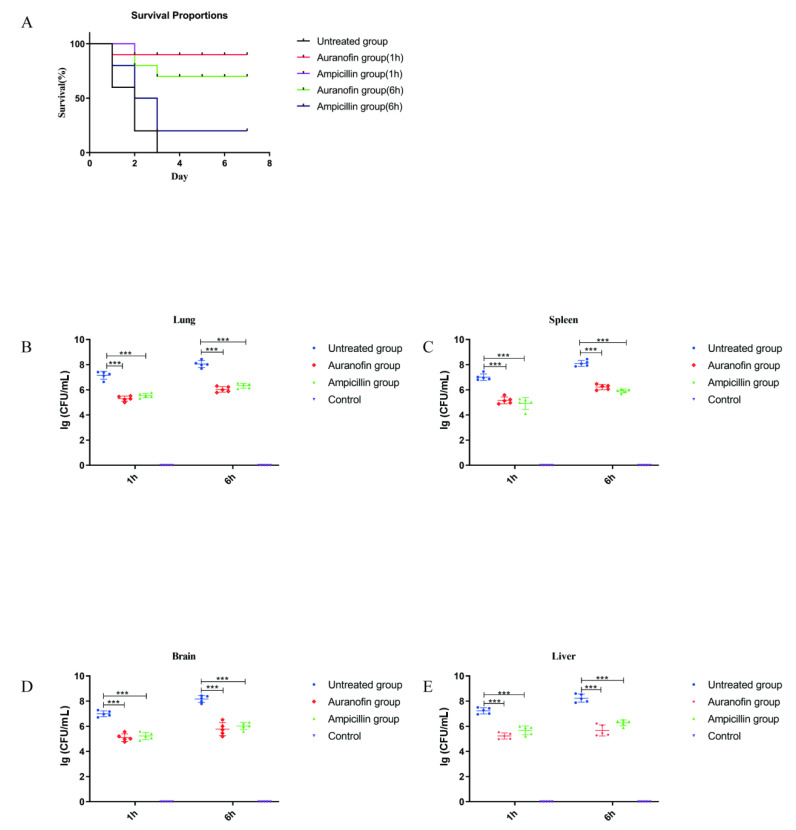
(**A**) The survival rates of severely infected mouse models treated with auranofin or ampicillin. The number of SC19 cells in the presence or absence of auranofin. Mice were intraperitoneally injected with 5 × 10^8^ CFU of SC19. The number of bacteria in the lung (**B**), spleen (**C**), brain (**D**) and liver (**E**) was determined at 8 h post infection (two-tailed, unpaired t-tests, *n* = 5). *** *p* < 0.001.

**Figure 7 antibiotics-10-00026-f007:**
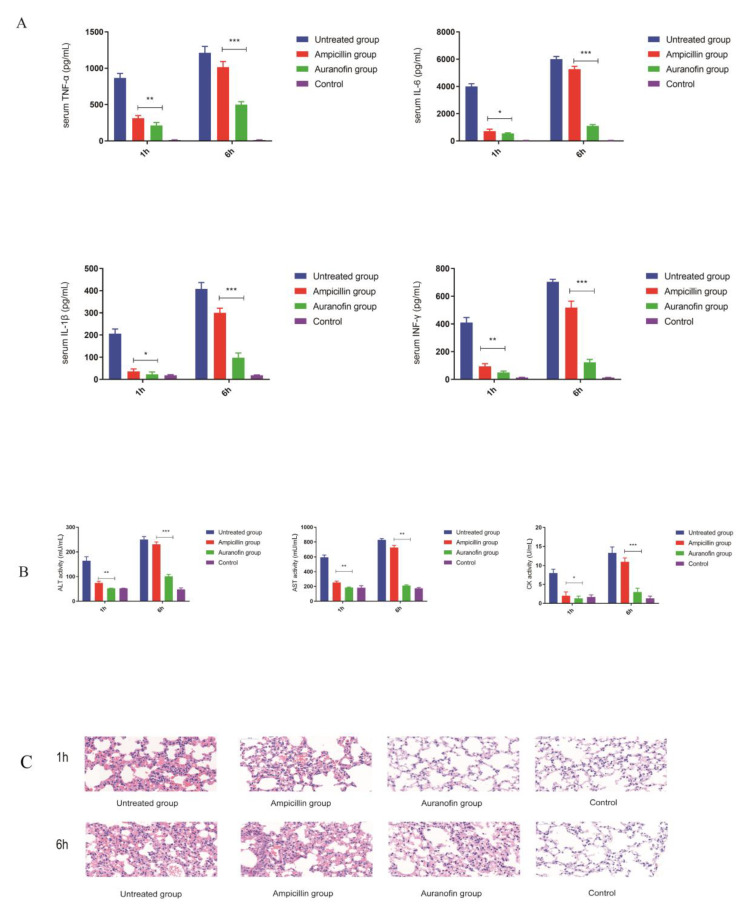
(**A**) Comparison of the anti-inflammatory activity of auranofin and ampicillin in SC19-infected mice. Serum levels of TNF-α, IL-6, IL-1β and INF-γ (two-tailed, unpaired t-tests, *n* = 5). (**B**) Comparison of the protective effect of auranofin and ampicillin on lung damage in infected mice. The H&E images (40×) of lung lesions after infection. (**C**) Blood levels of AST, ALT, and CK at 8 h treatment (two-tailed, unpaired t-tests, *n* = 5). * *p* < 0.05; ** *p* < 0.01; *** *p* < 0.001.

**Table 1 antibiotics-10-00026-t001:** Strain used in the study.

Strain ID	Phenotypic Properties	Source	Auranofin (mg/L)
SC19	Resistant to CLI, TET and LEV	China (Hu Bei)	0.25
*S. suis* (160413)	Resistant to CLI, TET, AMP and LEV	China (Hu Bei)	0.125
*S. suis* (16042)	Resistant to CLI, TET, AMP and LEV	China (Hu Nan)	0.25
*S. suis* (16091)	Resistant to TET, AMP and LEV and STX	China (Hu Bei)	0.0625
*S. suis* (16095)	Resistant to CLI, TET, AMP and STX	China (Guang Zhou)	0.125
*S. suis* (16072)	Resistant to CLI, TET, AMP, LEV and STX	China (Hu Bei)	0.125
*S. suis* (18051)	Resistant to CLI, TET, AMP and LEV	China (Hu Bei)	0.125
*S. suis* (180515)	Resistant to TET, AMP and LEV	China (Hu Bei)	0.0625
*S. suis* (170612)	Resistant to CLI, TET, AMP and STX	China (Hu Bei)	0.125
*S. suis* (170601)	Resistant to CLI, TET, AMP and LEV	China (Hu Bei)	0.25
*S. suis* (170603)	Resistant to TET, AMP, LEV and STX	China (Zhe Jiang)	0.125
*S. pneumoniae* (16035)	MDRSP	China (Shan Dong)	0.0625
*S. pneumoniae* (16076)	MDRSP	China (Shan Dong)	0.125
*S. agalactiae* (160205)	Beta-hemolytic, Serogroup: Group B	China (Shan Dong)	0.125
*S. agalactiae* (160503)	Beta-hemolytic, Serogroup: Group B	China (Shan Dong)	0.0625
*S**. aureus* (160206)	VRSA	China (Shan Dong)	0.125
*S**. aureus* (160408)	VRSA	China (Shan Dong)	0.125
pSET2-TrxB/SC19	overexpressed strain		1

Abbreviations: AMP, ampicillin; CTX, gentamicin; TET, tetracycline; LEV, levofloxacin; CLI, Lincomycin; STX, trimethoprim and sulphame-thoxazole.

**Table 2 antibiotics-10-00026-t002:** Liver and kidney functions in the blood of mice in the control and treated groups.

Treatment ^a^	ALT (U/L) ^b^	AST (U/L) ^b^	Creatinine (μmol/L)	Urea Nitrogen (mmol/L)
Control	42.52 ± 1.12	87.91 ± 1.78	44.12 ± 0.892	11.32 ± 1.68
Auranofin (2.4 mg/L)	41.21 ± 1.23 (*p* > 0.05)	87.31 ± 1.52 (*p* > 0.05)	44.34 ± 0.963 (*p* > 0.05)	12.52 ± 1.25 (*p* > 0.05)

^a^: Mice (*n* = 5 in each group) treated with PBS or the auranofin (2.4 mg/kg) once daily for three days. ^b^: ALT, alanine transaminase; AST, aspartate transaminase; U/L, international units per litre.

**Table 3 antibiotics-10-00026-t003:** List of oligonucleotide primers used in this study.

Primer	Sequence (5′-3′)	Remark
P1	CCCAAGCTATCCAGGCTATGACCATATTTCA(HindIII)	TrxB protein expression recombination vector
P2	CCGCTCCTATTCAGCTAGTTCTGTGATGTAG(XhoI)	
P3	CGCGGACTATCCAGGCTATGACCATATTTCA(BamHI)	TrxB overexpression vector
P4	CCGGAACTATTCAGCTAGTTCTGTGATGTAG(EcoRI)	

## Data Availability

The data presented in this study are available on request from the corresponding author.

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
