# Peer review of "Auranofin Has Advantages over First-Line Drugs in the Treatment of Severe Streptococcus suis Infections"

_antibiotics, 2020, doi:10.3390/antibiotics10010026_

Round 1
Reviewer 1 Report
Dear authors,
Overall comments
The authors have shown auranofin, an anti-rheumatic drug, as the potential treatment for Streptococcus suis infections. Authors have done in vitro toxicity tests, search for the binding site, and in vivo studies. Even though the authors conducted many kinds of experiments, the article has an inadequate background of the research and limited discussions of the results. Similar studies have been reported in the past few years, which were not cited in this article. The authors did not discuss the results by comparing them with previous studies nor connect the results from the experiments they have conducted (ex: cell injury protection, thiol-redox homeostasis, or STSLS).
In the mouse experiments, the authors have not shown whether the infection was established or if they have done similar pre-experiments before the real experiments. Even though the reviewed article is original, the in vivo experiments are very similar to that of Thangamani et al., 2016 (Scientific Reports). If the authors referred to the 2016 article, they should acknowledge and discuss further to compare the results.
Overall, the article contains too many errors in the figure labeling and orders. The article appreciates significant improvement in writing and presentation.
Comments per section
Abstract
More editing will help to clarify the abstract.
Line 31: decreasing the reducing capacity doesn’t make sense.
STSLS is shown twice.
No need to abbreviate MDRSP (MDR=multidrug resistant?).
Introduction
The introduction should include more background of auranofin and the use of auranofin for the treatments for bacterial infections citing the previous studies.
Results
- Section 2.2: Please add safety evaluation for RAW264.7 cells because these cells.
- Section 2.4: There is a possibility that the SC19 infected-dead cells were washed away with PBS before plating them on TSA plates (therefore more reduction of bacteria?). What is the control used in this experiment? Please add a figure of uninfected cells (without auranofin).
- Section 2.6: Did the authors count the intracellular bacteria in the enumeration?
Discussion
The discussion is not enough. Most of the contents of the discussion should be included in the introduction. The authors are not adequately discussing the results of the experiments. Please provide connections between the experiments to discuss the story of the mechanisms on how auranofin protects the host from STSLS. The authors included a discussion in the results section (ex. Section 2.7).
Methods
The order of the methods should follow that of the results.
References
Line 45: Please add reference No. 5 for the 1600 human cases.
Recheck if all the references are in order. It seems they are not matching.
Other comments
Line 51: Please spell out STSLS using for the first time in the main contents (Abstract does not count).
Line 65: What is SC19? Indicate that it is a strain name.
Line 72: Please give a background of TrxB protein.
Line 120~ : Please mention this is about mouse study.
Line 160-165: This should be the introduction.
Line 131: onice -> once
Line 160-161: Need to restructure the sentence.
Line 169: Did auranofin really “protect” the cells? Auranofin was added after the SC19 infection, which the macrophages seemed injured and dead already as shown in Figure 4A.
Line 187: Explain about pSET2-TrxB/SC19 cells
Line 191-196: These are discussion.
Abbreviation without spelling out (list of few but many more throughout the article)
Line 86: MDRSP, VRSA, MIC
LINE 89: MBC
Line 186: TrxR
Mixed use of TrxB and TxrB
Figures and tables
Figure 1: No x-scale label
Figure 2A and 2B should be reversed. Incorrect in Line 124.
Figure 2A: x-scale is not “post-infection”.
Figure 2B: No error bars for this one.
Figure 4: No scale bar.
Line 165-177: Explanation for Figure 4 is all wrong with wrong labeling.
Table 2: Treatmenta -> Treatment
Author Response
Response to Reviewer 1 Comments
Point 1:
Overall comments
The authors have shown auranofin, an anti-rheumatic drug, as the potential treatment for Streptococcus suis infections. Authors have done in vitro toxicity tests, search for the binding site, and in vivo studies. Even though the authors conducted many kinds of experiments, the article has an inadequate background of the research and limited discussions of the results. Similar studies have been reported in the past few years, which were not cited in this article. The authors did not discuss the results by comparing them with previous studies nor connect the results from the experiments they have conducted (ex: cell injury protection, thiol-redox homeostasis, or STSLS).
In the mouse experiments, the authors have not shown whether the infection was established or if they have done similar pre-experiments before the real experiments. Even though the reviewed article is original, the in vivo experiments are very similar to that of Thangamani et al., 2016 (Scientific Reports). If the authors referred to the 2016 article, they should acknowledge and discuss further to compare the results.
Overall, the article contains too many errors in the figure labeling and orders. The article appreciates significant improvement in writing and presentation.

Response 1:Thank you very much, we appreciate your thoroughly advices.
Point 2:
Comments per section
Abstract
More editing will help to clarify the abstract.
Line 31: decreasing the reducing capacity doesn’t make sense.
STSLS is shown twice.
No need to abbreviate MDRSP (MDR=multidrug resistant?).
Response 2: Thanks for the kind reminder. We have implemented your proposal and added description to clarify the abstract in revised manuscript. (Line35-36)
In line 31, we have revised the sentence: “Our results showed that auranofin can bind to the functional domain of bacterial thioredoxin reductase, decreasing the reducing redox-responsive capacity of target bacteria and allowing for the killing of S. suis cells.”(Line31)
We already deleted one of the shows about STSLS.(Line 26)
Yes, MDR=multidrug resistant. Also, we have corrected this omission: “......such as multidrug resistant Streptococcus pneumoniae (MDRSP).” (Line 33)
Point 3:
Introduction
The introduction should include more background of auranofin and the use of auranofin for the treatments for bacterial infections citing the previous studies.
Response 3: Thanks for your advice. We already added background of auranofin and the use of auranofin for the treatments for bacterial infections through citing the previous studies in introduction.(Line75-78)
Point 4:
Results
- Section 2.2: Please add safety evaluation for RAW264.7 cells because these cells.
- Section 2.4: There is a possibility that the SC19 infected-dead cells were washed away with PBS before plating them on TSA plates (therefore more reduction of bacteria?). What is the control used in this experiment? Please add a figure of uninfected cells (without auranofin).
- Section 2.6: Did the authors count the intracellular bacteria in the enumeration?
Response 4: We appreciate for your professional advices, and we addressed reviewer’s comments below.
- Section 2.2: We have added safety evaluation for RAW264.7 cells.(Line103-104)
2.The control group is uninfected group.We have added new illustrations to correct this. In this experiment, we used the same number of cells and bacteria, and the error bar was corrected by at least three repetitions.(Line 193-194)
- Section 2.6: In 2.6, we mainly researched the protection rates observed between the two drugs. We didn't count the intracellular bacteria in the enumeration.
Point 5:
Discussion
The discussion is not enough. Most of the contents of the discussion should be included in the introduction. The authors are not adequately discussing the results of the experiments. Please provide connections between the experiments to discuss the story of the mechanisms on how auranofin protects the host from STSLS. The authors included a discussion in the results section (ex. Section 2.7).
Response 5: Thank you for this kind reminder.We have added discussion carefully in the manuscript in accordance with your suggestion.(Line318-320)
Point 6:
Methods
The order of the methods should follow that of the results.
Response 6:Thank you very much for the suggestion.We have adjusted the order of the methods in accordance with order of the results.
Point 7:
References
Line 45: Please add reference No. 5 for the 1600 human cases.
Recheck if all the references are in order. It seems they are not matching.
Response 7: Thank you for this kind reminder. We have added reference No. 5 for the 1600 human cases,(Line 49) and rechecked carefully all the references in the manuscript.
Point 8:
Other comments
Line 51: Please spell out STSLS using for the first time in the main contents (Abstract does not count).
Line 65: What is SC19? Indicate that it is a strain name.
Line 72: Please give a background of TrxB protein.
Line 120~ : Please mention this is about mouse study.
Line 160-165: This should be the introduction.
Line 131: onice -> once
Line 160-161: Need to restructure the sentence.
Line 169: Did auranofin really “protect” the cells? Auranofin was added after the SC19 infection, which the macrophages seemed injured and dead already as shown in Figure 4A.
Line 187: Explain about pSET2-TrxB/SC19 cells
Line 191-196: These are discussion.
Response 8: Thanks the reviewer for his thoroughly comment. We addressed reviewer’s comments below.
Line 51: We have spelt out STSLS using for the first time in the main contents: “......where Streptococcal toxic shock-like syndrome (STSLS)......” (Line 54-55)
Line 65: Yes, SC19 is a strain name, and we have supplemented the specific name:“ ......S. suis 2 strain SC19.......” in our manuscript.(Line 69)
Line 72: We have added the background of TrxB protein.(Line 74-75)
Line 120~ : To avoid misunderstanding, we have made the following changes: “At an auranofin concentration of 32 µg/ml, the viability rate of Vero cell was 50%, revealing that the toxicity of auranofin toward cells was much higher than the MIC against S. suis (Figure 2B).”(Line 133-144)
Line 160-165: The reason why we put those sentences on here is we want readers understand the importance of intracellular bacteria treatment. Through reference the article Mechanistic Understanding Enables the Rational Design of Salicylanilide Combination, Therapies for Gram-Negative Infections,Metformin Restores Tetracyclines Susceptibility against Multidrug Resistant Bacteria.(Line 177-182)
Line 131: We have changed onice to once.(Line 147)
Line 160-161: We have restructured the sentence: “Although S. suis is an atypical intracellular bacterium, the SS2 virulence factor SLY has been reported to promote host cell perforation”.(Line177-1)
Line 169: Yes, auranofin really protected the cells. We are sorry for our negligence in writing the article, so that the reviewer has misunderstood it. Here,we have corrected our errors.
Line 187:We have explained about pSET2-TrxB/SC19 cells in materials and methods.“The TrxB overexpression vector pSET2-TrxB/SC19 was constructed using the Escherichia coli-Streptococcus suis shuttle vector pSET2”.(Line 313-315)
Line 191-196: Also, we write those sentences here in order to let readers understand the importance of intracellular bacteria treatment. And this description way is quoted from Mechanistic Understanding Enables the Rational Design of Salicylanilide Combination Therapies for Gram-Negative Infections,Metformin Restores Tetracyclines Susceptibility against Multidrug Resistant Bacteria.(Line211-218)
Point 9:
Abbreviation without spelling out (list of few but many more throughout the article)
Line 86: MDRSP, VRSA, MIC
LINE 89: MBC
Line 186: TrxR Mixed use of TrxB and TxrB
Response 9:Thank you for this kind reminder.We have spelt out all of the abbreviations in this article.
Line 86: “multidrug resistant S. pneumoniae (MDRSP), vancomycin-resistant S. aureus (VRSA), minimum inhibitory concentration (MIC)”
LINE 89: “minimum bactericidal concentration (MBC)”
Line 186: It’s a spelling mistake, TrxR is TrxB.
Point 10:
Figures and tables
Figure 1: No x-scale label
Figure 2A and 2B should be reversed. Incorrect in Line 124.
Figure 2A: x-scale is not “post-infection”.
Figure 2B: No error bars for this one.
Figure 4: No scale bar.
Line 165-177: Explanation for Figure 4 is all wrong with wrong labeling.
Table 2: Treatmenta -> Treatment
Response 10: Thanks the referee for taking time to assess our manuscript, and we apologize for those mistakes.
Figure 1: We have added x-scale label.
Figure 2A and 2B have been reversed.
Figure 2A: X-scale has been revised (Figure 2B).
Figure 2B:Error bars is not obvious because of the good repeatability of different datas.
Figure 4: We have added scale bar in figure note.
Line 165-177: We are sorry for our negligence in writing, so that the reviewer has misunderstood it. We have corrected all errors in figure 4.
Table 2: We have changed “Treatmenta” to “Treatment”.

Reviewer 2 Report
The manuscript titled “Auranofin has advantages over first-line drugs in the treatment of severe Streptococcus suis infections” describes an in-depth study about auranofin and its antibacterial properties against S. suis. The manuscript is well-written; the aim of the work is clear, and authors’ comments agree with the results. Indeed, after the evaluation of auranofin as antibacterial agent (against some Staphylococcal and Streptococcal strains), the authors investigated the mechanism of action of auranofin, also proving the binding with TxrB by ITC. In parallel, toxicity evaluation of auranofin on Vero cells and also in vivo highlighted a good safety profile for the compound, which also showed a promising antibacterial activity in in vivo experiments.
Some minor issues are present:
- Page 2, line 81: please specify that pathogens are only Staphylococcal and Streptococcal strains.
- Page 4, line 120: please introduce in vivo experiments before discussing liver and kidney blood indices of urea nitrogen.
- Page 7, line 160: it seems there is a problem in the sentence.
- Page 14, line 299: replace “phenomenon.” with “phenomenon;”.
Author Response
Response to Reviewer 2 Comments
Point 1:
The manuscript titled“Auranofin has advantages over first-line drugs in the treatment of severe Streptococcus suis infections”describes an in-depth study about auranofin and its antibacterial properties against S. suis. The manuscript is well-written; the aim of the work is clear, and authors’ comments agree with the results. Indeed, after the evaluation of auranofin as antibacterial agent (against some Staphylococcal and Streptococcal strains), the authors investigated the mechanism of action of auranofin, also proving the binding with TxrB by ITC. In parallel, toxicity evaluation of auranofin on Vero cells and also in vivo highlighted a good safety profile for the compound, which also showed a promising antibacterial activity in in vivo experiments.
Response 1:We appreciate for your positive comments.
Point 2:
Page 2, line 81: please specify that pathogens are only Staphylococcal and Streptococcal strains.
Response 2:Thanks for your suggestion. We have made the following changes: “To assess the antimicrobial activity of auranofin, we evaluated the effect on auranofin against 18 multidrug-resistant Staphylococcal and Streptococcal strains (Table 1).” (Line 89-90)
Point 3:
Page 4, line 120: please introduce in vivo experiments before discussing liver and kidney blood indices of urea nitrogen.
Response 3: Thanks for your kind reminder. We have added this part in materials and methods.(Line 362-370)
Point 4:
Page 7, line 160: it seems there is a problem in the sentence.
Response 4: We have revised the sentence: “Although S. suis is an atypical intracellular bacterium,the SS2 virulence factor SLY has been reported to promote host cell perforation”.(Line 177-178)
Point 5:
Page 14, line 299: replace “phenomenon.” with “phenomenon;”.
Response 5: Thanks for your proposal. We have replace “phenomenon.” with “phenomenon;”.(Line320 )

Reviewer 3 Report
Comments to the author:
This paper assesses the advantage of Auranofin in the treatment of Stretococcus suis infection over first-line drugs like Ampicillin. The result presented in the manuscript showed that auranofin bind to the functional domain of TxrB causing depletion of thiols and thus reducing the defense against oxidative stress. Moreover, auranofin has an anti-bacterial effect on other Gram-positive organisms and reduce the mortality of mice by decreasing the infections by S. suis. Based on their observations, the authors concluded that auranofin can act as a potential antibiotic alone or in combination against S. suis. Overall, this is a very well written manuscript that clearly shows the importance of Auranofin as an antibiotic against S. suis. The data presented are clear, easy to understand (few exception-see comments below), method sections are well described. However, there are some editorial points/comments/questions that the authors are trusted to address before further consideration (line numbers correspond to those in the uploaded in the manuscript).
Major comments:
- Line 141-143, Based on the molecular docking author found that Auranofin stretched into the hydrophobic pocket of the TrxB binding site that consisted of Cys-130, Ala-131, Val-132, Phe-158, and Leu-279, forming a strong hydrophobic structure. My suggestion will be to carry out mutagenesis on these residues check the effect using Auranofin and TrxB Binding Assays. Moreover, It will be interesting to know the overexpression of these variants on MICs as compared to wild-type. This is an important experiment needed to support the molecular docking data.
- Did the author try to make ∆trxB strain and check the effect of Auranofin on it?
Minor points:
- The manuscript needs careful proofreading. The scientific name should be italicized throughout. (example Line 86).
- In figure 1. Label X-axis and correct the figure labels as 4X MIC.
- In figure 3, the panel B label should be enlarged.
- Figure 4, the font size is inconsistent. For panel C, include the lower dose image of SC19-infected cells treated with less than 0.25 mg/l of auranofin.
- Figure 6, all the panel images should be constant. Make changes to B-E.
- Figure 7, Panel C is very hard to analyze. Enlarge them.
Author Response
Response to Reviewer 3 Comments
Point 1:
This paper assesses the advantage of Auranofin in the treatment of Stretococcus suis infection over first-line drugs like Ampicillin. The result presented in the manuscript showed that auranofin bind to the functional domain of TxrB causing depletion of thiols and thus reducing the defense against oxidative stress. Moreover, auranofin has an anti-bacterial effect on other Gram-positive organisms and reduce the mortality of mice by decreasing the infections by S. suis. Based on their observations, the authors concluded that auranofin can act as a potential antibiotic alone or in combination against S. suis. Overall, this is a very well written manuscript that clearly shows the importance of Auranofin as an antibiotic against S. suis. The data presented are clear, easy to understand (few exception-see comments below), method sections are well described. However, there are some editorial points/comments/questions that the authors are trusted to address before further consideration (line numbers correspond to those in the uploaded in the manuscript).
Response 1:We appreciate for your detailed review and comments.
Point 2:
Line 141-143, Based on the molecular docking author found that Auranofin stretched into the hydrophobic pocket of the TrxB binding site that consisted of Cys-130, Ala-131, Val-132, Phe-158, and Leu-279, forming a strong hydrophobic structure. My suggestion will be to carry out mutagenesis on these residues check the effect using Auranofin and TrxB Binding Assays. Moreover, It will be interesting to know the overexpression of these variants on MICs as compared to wild-type. This is an important experiment needed to support the molecular docking data.
Response 2:Thanks again for your grate recommendation. In this paper, we mainly found auranofin inhibit the activity of TrxB protein. Firstly, we simulated the binding of auranofin and TrxB through molecular docking. Secondly, we verified the binding of the auranofin and TrxB through ITC experiments. Thirdly, we also verified that auranofin can inhibit the activity of TrxB through phenotypic experiments in vitro. Therefore, amino acid residues are not the focus of our forward research. However, the reviewer gave us a good suggestion, which provided a good direction for our further research.
Point 3: Did the author try to make ∆trxB strain and check the effect of Auranofin on it?
Response 3: Yes, we did. But we found that the bacteria died because of the loss of the trxB, which indicated TrxB is an indispensable protein in the life of bacteria.
Point 4: The manuscript needs careful proofreading. The scientific name should be italicized throughout. (example Line 86).
Response 4:Thanks for your advice. We have proofread our manuscript and reversed the scientific name throughout.
Point 5: In figure 1. Label X-axis and correct the figure labels as 4X MIC.
Response 5:Thank you for this kind reminder. We have corrected the figure 1 as you mentioned above.
Point 6: In figure 3, the panel B label should be enlarged.
Response 6:Thank you very much for the suggestion. And this is a high-resolution picture that will be very clear after zooming in.
Point 7: Figure 4, the font size is inconsistent. For panel C, include the lower dose image of SC19-infected cells treated with less than 0.25 mg/l of auranofin.
Response 7:Thank you very much for the reminder. We have adjusted the font size to keep it consistent in Figure 4.
Point 8: Figure 6, all the panel images should be constant. Make changes to B-E.
Response 8: We appreciate this kind remind. We already revised Figure 6.
Point 9: Figure 7, Panel C is very hard to analyze. Enlarge them.
Response 9: We appreciate this valuable advice. We have enlarged panel C in figure 7.
Round 2
Reviewer 3 Report
Kindly find my comments as an attachment.

Author Response
Reviewer: This is very important information and needs to be included in details in the paper including the methodology used by the author for making the ∆trxB strain. I regret to inform the response provided by the author is not satisfactory and therefore more details are needed as there are reports in the literature which suggest ∆trxB strains are viable in Streptococcus species (PMID: 25280752). How did the author confirm that trxB is indispensable (provide evidence and details)? Did they try to put the complementing plasmid carrying trxB into the strain and then whether they were successful in carrying out deletion and confirmed through PCR later on? All these questions need to be answered and further evidence are needed to support the authors claim.
Response 1: We appreciate for your professional question. We used S.suis knockout system (PMID: 11322824), and constructed a gene mutant library(PMID: 26946377),(PMID: 30105012). Such as cps2E, cps2G, cps2J, cps2L, Sly, et al. The method was described as previously reported(PMID: 26946377). As Michael B. Harbutet showed, the Trx-TrxR system is often essential in these GSH-lacking Gram-positive bacteria organisms, and they didn't get trxB2 knockout strain. We noticed the article used S. Gordonii (PMID: 25280752) as a research object. This might because of the difference between S.suis. and S. Gordonii. Besides, we constructed TrxB overexpression strain pSET2-TrxB/SC19, and the resistance of Auranofin increased significantly. We thought those dates could improve that TrxB is a auranofin target of S.suis.
Reviewer: Figure 4, For panel C, include the lower dose image of SC19-infected cells treated with less than 0.25 mg/l of auranofin. I guess the author overlooked the comment and. Kindly include the image with a lower dose image.
Response 2:We are very sorry for our negligence. Many thanks for your reminders, but we here did only the treatment with 0.25mg/L auranofin and did not do the treatment with a low dose.(line 430)